

# The impact of spring wheat species and sowing density on soil biochemical properties, content of secondary plant metabolites and the presence of *Oulema* ssp.

Jarosław Pobereżny[1], Elżbieta Wszelaczyńska[1], Robert Lamparski[2], Joanna Lemanowicz[3], Agata Bartkowiak[3], Małgorzata Szczepanek[4] and Katarzyna Gościnna[1]

[1] Institute of Agri-Foodstuff Commodity/Faculty of Agriculture and Biotechnology, Bydgoszcz University of Science and Technology, Bydgoszcz, Poland
[2] Department of Biology and Plant Protection/Faculty of Agriculture and Biotechnology, Bydgoszcz University of Science and Technology, Bydgoszcz, Poland
[3] Department of Biogeochemistry and Soil Science/Faculty of Agriculture and Biotechnology, Bydgoszcz University of Science and Technology, Bydgoszcz, Poland
[4] Department of Agronomy/Faculty of Agriculture and Biotechnology, Bydgoszcz University of Science and Technology, Bydgoszcz, Poland

Corresponding author
Jarosław Pobereżny,
poberezny@pbs.edu.pl

## ABSTRACT

The physical and chemical properties of the soil are important factors influencing the yield of crops. One of the agrotechnical factors influencing the biochemical properties of soil is sowing density. It affects the yield components, light, moisture and thermal conditions in the canopy and the pressure of pests. Secondary metabolites, many of which are known to act as a defense mechanism against insects, are of importance in the interaction between the crop and abiotic and biotic factors of the habitat. To the best of our knowledge, the studies conducted so far do not sufficiently reveal the impacts of the wheat species and the sowing density, together with the biochemical properties of the soil, on the accumulation of bioactive ingredients in the crop plants, and the subsequent impacts on the occurrence of phytophagic entomofauna in various management systems. Explaining these processes creates an opportunity for more sustainable development of agriculture. The study aimed to determine the effect of wheat species and sowing density on the biochemical properties of the soil, concentrations of biologically active compounds in the plant and the occurrence of insect pests in organic (OPS) and conventional (CPS) production systems. The research was conducted on spring wheat species (Indian dwarf wheat—*Triticum sphaerococcum Percival* and Persian wheat—*Triticum persicum Vavilov*) grown in OPS and CPS at sowing densities 400, 500, 600 (seeds m$^{-2}$). The following analyzes were performed: (i) soil analysis: the activity of catalases (CAT), dehydrogenases (DEH), peroxidases (PER); (ii) plant analysis: total phenolic compounds (TP), chlorogenic acid (CA), antioxidant capacity (FRAP); (iii) entomological analysis of the number of insects—*Oulema* spp. adults and larvae. Performing analyzes in such a wide (interdisciplinary) scope will allow for a comprehensive understanding of the soil-plant-insect biological transformation evaluation. Our results showed that an increase in soil enzyme activity caused a decrease

in TP contents in the wheat grown the OPS. Despite this, both the content of TP and the anti-oxidative activity of the ferric reducing ability of plasma (FRAP) were higher in these wheats. Bioactive compound contents and FRAP were most favoured by the lowest sowing density. Regardless of the production system, the occurrence of the *Oulema* spp. adults on *T. sphaerococcum* was the lowest at a sowing density of 500 seeds m$^{-2}$. The occurrence of this pest's larvae was lowest at a sowing density of 400 seeds m$^{-2}$. Research on bioactive compounds in plants, biochemical properties of soil and the occurrence of pests make it possible to comprehensively assess the impact of the sowing density of ancient wheat in the ecological and conventional production system, which is necessary for the development of environmentally sustainable agriculture.

# INTRODUCTION

Soil is not inanimate, but is the richest of natural systems in terms of species composition. The physical and chemical properties of soil are an important factor affecting crop yields. Granulometric composition is one of the most basic soil features determining its sorption capacity, buffering capacity and water retention, and the bioavailability of elements for plants (*Moral & Rebollo, 2017*). Soil enzymes catalyse the processing of matter and energy (*Burns et al., 2013*). They are involved in breaking down organic matter and making mineral substances available to plants (*Zhao et al., 2016*). Enzymatic activity is thus defined as a measure of soil fertility and productivity, which determine yield and quality (*Chu, Zaid & Eivazi, 2016*; *Piotrowska-Długosz, 2019*). Oxidoreductases are a class of enzymes that CAT the transfer of electrons from donor to acceptor. The acceptor can be an organic and inorganic compound, or oxygen (*Strek & Telesinski, 2017*). The DEH activity is an indicator of the intensity of the respiratory metabolism of all soil microorganism populations, and is used to determine the total microbial activity of the soil (*Furtak & Gajda, 2017*; *Lemanowicz et al., 2020a*; *Lemanowicz et al., 2020b*). The enzymes CAT and PER mediate key processes in the soil ecosystem, such as lignin degradation, humification and mineralisation (*Baldrian & Šnajdr, 2010*). The basic agrotechnical factor influencing the biochemical properties of the soil is plant density (*Hinsinger, 2001*). Regulation of plant growth characteristics and seeding density affects light, humidity and thermal conditions as well as pathogen pressure in the crown of crops (*Postma et al., 2020*). It also affects aspects of yield, in particular ear density per unit of area (*Szczepanek et al., 2020*). Excessive plant density causes plants to compete for soil resources and become susceptible to disease. Conversely, excessively sparse sowing reduces the spread of diseases, but may increase weed pressure (*Kheshtzar & Siadat, 2015*; *Lemanowicz et al., 2020a*; *Lemanowicz et al., 2020b*; *Szczepanek et al., 2020*).

In the interaction between crops and abiotic and biotic habitat factors, secondary metabolites are of particular importance. They are estimated to number about 15,000 in specific plant species (*Fernie, 2007*). They are not involved in the primary metabolism, but

are essential in plants' adaptation to changing environmental conditions (*Howe & Jander, 2008*). Secondary metabolites include TP, and particularly phenolic acids (*Paszkiewicz et al., 2012*; *D'Archivio et al., 2007*; *Ullah & Khan, 2008*), which regulate the course of defensive reactions to pathogenic attacks (*Kusano et al., 2007*; *Raju, Jayalakhsmi & Sreeramulu, 2009*; *Heil, 2009*; *Moloi & Van der Westhuisen, 2009*). Some act as a defence mechanism against herbivores, making plant tissues less palatable. They also protect plants against microorganisms and competing plants (*War & War, 2011*; *Barakat et al., 2010*). One theory is that TP contents in plants depend on environmental conditions (*Mpofu, Sapirstein & Beta, 2006*; *Przybylska-Balcerek, Frankowski & Stuper-Szablewska, 2020*). Plant TP contents rise because of plants' defensive reaction to stress conditions such as ultraviolet radiation or attack by pathogens (*War & War, 2011*; *Usha Rani & Jyothsna, 2010*; *Sharma, Sujana & Rao, 2009*; *Pandey & Rizvi, 2009*). The concentrations of phenols in plants may also depend on development stage (*Manach et al., 2004*) and number of shoots (*Stumpf, Yan & Honermeier, 2019*). In order to reduce pest-related quantitative and qualitative losses in grain yield, it is necessary to constantly monitor crops to identify perpetrators and assess the extent of damage. Prophylactic action is important to reduce the risk of pests proliferating (*Kaniuczak & Bereś, 2011*). Important means for reducing risk include selection of varieties, site selection, balanced fertilisation, but also the proper sowing density and plant density (*Szczepanek et al., 2020*; *Kaniuczak & Bereś, 2011*).

There is currently insufficient knowledge on how abiotic and biotic environmental factors interact with crops' metabolic reaction under changing agrotechnical conditions. The studies conducted so far do not sufficiently explain the influence that the quality of primary wheat species in terms of bioactive ingredients have on the occurrence of insects feeding on CPS and OPS cultivated crops.

Elucidating these processes will create an opportunity for the sustainable agricultural development by reintroducing the original species of cereals and increasing biodiversity in ecosystems. Therefore, research was undertaken to determine the effect of wheat species and sowing densities on the biochemical properties of the soil, concentrations of biologically active compounds in crops and the occurrence of insect pests depending on the OPS and CPS systems.

## MATERIAL & METHODS

### Site description and crop management

Field tests and soil, plant and entomological sampling were carried out in 2018–2020, at the beginning of June in the flag leaf stage of spring wheat (BBCH 39). The research was conducted on spring wheat grown in OPS and CPS. The experimental factors were: factor Ix—spring wheat species (Fig. 1); Indian dwarf wheat (*Triticum sphaerococcum* Percival) and Persian wheat (*Triticum persicum* Vavilov); factor II—sowing density 400, 500, 600 (seeds m). The experiments were performed in a split-plot arrangement in four replicates. The single plot size of 21 m². *Triticum sphaerococcum* Percival cv. Trispa and *Triticum persicum* Vavilov cv. Persa used in our field experiments are the cultivars, which has been bred by the Bydgoszcz University of Science and Technology in 2020. The Breeder's Right,

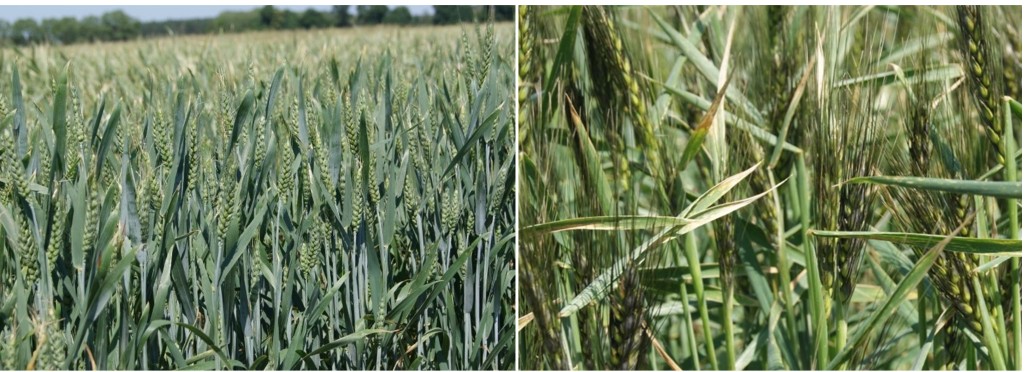

**Figure 1  Indian dwarf wheat (left) and Persian wheat (right).**

granted by the director of Research Centre for Cultivar Testing, is exercisable on the territory of Poland at the national Plant Breeders' Rights protection level. Details on soil characteristics prior to the experiments can be found in *Lemanowicz et al. (2020a)* and *Lemanowicz et al. (2020b)*. The forecrops for the tested species of OPS and CPS cultivated spring wheats were cereals (triticale or winter wheat). Immediately after harvesting the forecrop, an intercrop of peas of the tendril-leaved variety 'Tarchalska' were sown at a rate of 240 kg ha[1]. Pre-winter ploughing was carried out to a depth of 0.22–0.23 m. Sowing parameters, as well as methods of fertilization and plant protection against weeds, diseases and pests were provided by *Lemanowicz et al. (2020a)* and *Lemanowicz et al. (2020b)*.

## Soil analysis

Air-dried disturbed soil samples were sieved through a ø2-mm mesh and the selected physicochemical properties in soil were determined as follows: the content of clay fraction composition by laser diffraction method using a Masterssizer MS 2000 analyser (Malvern Panalytical, Worchestershire, UK); the soil pH in $CaCl_2$ (0.01 M) was measured potentiometrically (*PN-ISO, 1997*); organic carbon (OC) was measured using a Skalar TOC Primacs analyser (Skalar, The Netherlands). Fresh soils enzymatic activities were determined on soil samples that had been stored at 4 °C for no more than two weeks. The activity of three redoxidoreductase enzymes was determined. The activity of DEH (E.C.1.1.1) and CAT (E.C.1.11.1.6) in soil was determined by methods described by *Bartkowiak, Breza-Boruta & Lemanowicz (2016)*. Analyses of (PER) activity (EC 1.11.1.7) were carried out using the Bartha and Bordeleau method (*Bartha & Bordeleau, 1969*) by measuring the amount of purpurogallin (PPG) formed as a result of the oxidation of pyrogallol in the presence of $H_2O_2$. The absorbance of the solution was measured colorimetrically at $\lambda = 460$ nm using a spectrophotometer.

Based on the enzymatic activities of the samples, the geometric mean of enzyme activities (G*Mea*) was calculated using a method (*Hinojosa et al., 2004*) as follows:

$$GMea = \sqrt[3]{(CAT * DEH * PER)} \qquad (1)$$

where: CAT, DEH, PER are referred to catalase, dehydrogenase and peroxidase respectively.
## Plant analysis

Fifty generative tillers were randomly collected from each plot and used for analysis.

## Samples preparation

Plant samples were collected for laboratory analyses in individual experimental plots. Five consecutive stalks (which were cut with a sharp knife above the soil) were taken completely randomly in five places. The samples were put in a plastic bag, labelled immediately upon arrival from the field. The tillers were incubated in the laboratory storage at 5 °C and 90% RH. After 3 days the samples were cut and well mixed. A representative from each sample was frozen and stored at least at −20 °C in a freezer (Whirpool AFG 6402 E-B, Benton Harbor, MI, USA). For laboratory analyses the frozen material was freeze-dried (Alpha model 1–4 LSC, Martin Christ, Osterode, Germany). The materials were lyophilized to permanent weight with a moisture content <2%. Dried wheats were also milled to obtain a fine powder (particle size 0.3–0.5 mm mesh) using an Ultra-Centrifuge Retsch mill ZM 100 (Retsch, Germany). The ground samples were stored in the dark in air-tight bags in desiccators.

## The TP content analysis

The determination of TP content was conducted using the method developed by *Fang et al. (2006)*, which includes the measurement of absorbance with the usage of Folin–Ciocalteu reagent. The assay was conducted as follows: 200 μL of a suitably diluted sample (1/10 lyophilized sample/methanol) was transferred to a test tube and 800 μL of dionized water was added. After thorough mixing, five mL of 0.2 N Folin–Ciocalteu reagent was added to the test tube and vortexed. After 3 min of incubation at room temperature, 4 mL of sodium carbonate was added (75 g $L^1$). The prepared samples were then incubated in a dark room for two hours. Absorbance was measured (Shimadzu UV-1800, UV Spectrophotometer System, Kyoto, Japan) at the layer thickness of one cm and at a wavelength of 735 nm. The TP was calculated based on the standard curve prepared from chlorogenic acid.

## The CA content analysis

The CA content was determined colorimetrically by the method of *Griffiths, Bain & Dale (1992)*. Briefly, freeze-dried wheat powder (100 mg) was suspended in a two mL of urea solution (0.17 M) and acetic acid (0.10 M). One mL of sodium nitrite (0.14 M) was added to this solution and incubated for 2 min at room temperature before adding one mL of sodium hydroxide (0.5 M). The reaction was then centrifuged (Hettina Zentrifugen, Rotina 420 R, Tuttlingen, Germany) at 2250 g for 10 min and the absorbance of the cherry red complex formed of the supernatant was read at 510 nm (Shimadzu UV-1800, UV Spectrophotometer System, Kyoto, Japan). A standard curve was prepared using different concentrations of CA.

## Evaluation of the antioxidant capacity (FRAP method)

Antioxidant capacity was determined by FRAP (ferric reducing antioxidant power assay) method as described by *Benzie & Strain (1996)*. A total of 250 mL of acetate buffer (POCH, Gliwice) at pH 3.6 was mixed with 25 mL of TPTZ-2,4,6-tripyridyl-s-triazine solution

(Sigma–Aldrich, St. Louis, MO, USA) and 25 ml of iron (III) chloride hexahydrate solution (Chempur, Piekary Śląskie, Poland). The reaction mixture was prepared immediately prior to assay. The solution was incubated at 37 °C, throughout the analysis. A total of 6 mL of reaction mixture was mixed with 200 µL of sample and 600 µL of $H_2O$. The absorbance of the mixture was measured at 593 nm (Shimadzu UV-1800, UV Spectrophotometer System, Kyoto, Japan) after 4 minutes. Acetate buffer was used as a control. The control sample absorbance value was subtracted from the test sample values. Based on the performed measurements, absorbance values' dependence on sample concentrations was plotted. That was used to determine the absorbance value at the mean concentration of the applied dilutions, and anti-oxidative capacity was calculated at the same absorbance using the standard curve determined for $Fe^{2+}$ ions.

### Entomological analysis

Insect abundances of *Oulema* spp. adults and larvae were assessed after insects were collected by entomological net (*Tratwal et al., 2015*; *Lamparski, 2016*). Observations were made in the flag leaf stage of spring wheat, *i.e.*, BBCH (*Hinojosa et al., 2004*). Cereal leaf beetle adults and larvae abundances were made in four replicates. The results are presented in specimens per plot.

### Statistical analysis

The three-year research results were statistically verified by analysis of variance. The significance of differences was evaluated using the Tukey multiple confidence intervals for the significance level of $\alpha = 0.05$. The analysis of data variance was calculated using Statistica® software, and the main effects were tested by ANOVA. To evaluate the dependence between soil compositions and chemicals compounds in the plants the correlation coefficients (linear Pearsons correlation) was determined. All laboratory tests were carried out with three replications. Averages of results as well as standard deviation are shown in tables.

## RESULTS

### Enzymatic activity and soil properties

Granulometric composition analysis of the soil samples showed that the clay fraction was low. It ranged from 3.77 to 4.59% for OPS and from 4.70 to 5.48% for CPS (Table 1). According to the USDA (*USDA, 2006*) classification, all tested soil samples belonged to a single granulometric group–sandy loam. The soils were characterised by pH in 0.01 M $CaCl_2$, with values ranging in OPS from 5.02 to 6.08 (which is characterised as acid and slightly acid soil) and in CPS from 7.07 to 7.22 (neutral and alkaline soil).

Only the species of primary wheat grown in OPS significantly influenced soil OC content (Table 2). The highest accumulation of this macro-element (0.735%) was obtained in the soil with a plant density of 500 seeds $m^2$ of *T. persicum*. The cultivation of *T. persicum* contributed to a 10% increase OC content over *T. sphaerococcum*. Sowing density significantly ($p \leq 0.05$) affected OC content. In OPS, higher OC content was obtained (0.699% on average for the two wheats) using a sowing density of 400 seeds $m^2$. By

**Table 1  The content of clay fraction (%) and soil pH.**

| Sowing density (seeds m$^{-2}$) | OPS | | | | CPS | | | |
|---|---|---|---|---|---|---|---|---|
| | *T. sphaerococcum* | | *T. persicum* | | *T. sphaerococcum* | | *T. persicum* | |
| | Clay (%) | pH CaCl$_2$ | Clay (%) | pH CaCl$_2$ | Clay (%) | pH CaCl$_2$ | Clay (%) | pH CaCl$_2$ |
| 400 | 4.41 ± 0.07 | 5.02 ± 0.01 | 4.29 ± 0.06 | 5.39 ± 0.08 | 5.26 ± 0.08 | 7.04 ± 0.01 | 5.11 ± 0.01 | 7.20 ± 0.07 |
| 500 | 4.33 ± 0.05 | 5.34 ± 0.03 | 4.33 ± 0.05 | 5.71 ± 0.02 | 5.47 ± 0.03 | 7.12 ± 0.02 | 4.91 ± 0.01 | 7.22 ± 0.02 |
| 600 | 4.59 ± 0.03 | 5.52 ± 0.03 | 4.33 ± 0.01 | 6.08 ± 0.03 | 5.70 ± 0.05 | 7.14 ± 0.03 | 5.70 ± 0.04 | 7.18 ± 0.01 |
| LSD$_{0.05}$ | Clay: A − 0.034; B − 0.09; A/B − 0.091; B/A − 0.127 | | | | Clay: A − 0.083; B − 0.077; A/B − 0.108; B/A − 0.109 | | | |
| | pH CaCl$_2$: A − 0.083; B − 0.095; A/B − 0.116; B/A − 0.135 | | | | pH CaCl$_2$: A − 0.063; B − ns*; A/B − ns; B/A − ns | | | |

**Table 2  The content of soil organic carbon (OC%).**

| Sowing density (seeds m$^{-2}$) [B] | OPS | | | CPS | | |
|---|---|---|---|---|---|---|
| | Spring wheat species [A] | | | | | |
| | *T. sphaerococcum* | *T. persicum* | mean | *T. sphaerococcum* | *T. persicum* | mean |
| | OC (%) | | | | | |
| 400 | 0.703 ± 0.002 | 0.696 ± 0.004 | 0.699 ± 0.004 | 0.726 ± 0.003 | 0.705 ± 0.004 | 0.716 ± 0.011 |
| 500 | 0.566 ± 0.006 | 0.735 ± 0.005 | 0.650 ± 0.084 | 0.873 ± 0.002 | 0.825 ± 0.004 | 0.849 ± 0.024 |
| 600 | 0.536 ± 0.021 | 0.583 ± 0.005 | 0.560 ± 0.024 | 0.757 ± 0.002 | 0.884 ± 0.003 | 0.821 ± 0.064 |
| mean | 0.602 ± 0.073 | 0.671 ± 0.064 | 0.636 ± 0.035 | 0.785 ± 0.063 | 0.805 ± 0.074 | 0.795 ± 0.010 |
| | LSD$_{0.05}$ A − 0.004; B − 0.011; A/B − 0.011; B/A − 0.016 | | | LSD$_{0.05}$ A − ns*; B − 0.012; A/B − 0.021; B/A − 0.017 | | |

**Notes.**
*ns - not significant.

contrast, in CPS, 600 seeds m$^2$ resulted in the highest accumulation of OC in soil samples (0.821% on average for the two wheats). The OC content for the cultivation of ancient wheat species was higher in CPS than OPS. Based on the European Soil Database (*Gonet, 2007*) classes of OC content in soils, the studied soils are classified as having a very low OC content (<1%). It is assumed that OC = 2% is a critical level and that soils with a lower carbon content need to be exploited using methods (including such agrotechnical measures) that will increase their organic matter resources.

Variance analysis established that the applied experiment factors significantly influenced ($p \leq 0.05$) the activity of enzymes from the oxidoreductase class (Table 3). However, this effect varied depending on the enzyme tested. The cultivated species of spring wheat significantly changed CAT activity, but only in soil samples from CPS. The CAT activity was significantly higher in the soil hosting *T. sphaerococcum* (average 0.840 mg H$_2$O$_2$ kg$^1$ h$^1$) than in that from *T. persicum* (average 0.809 mg H$_2$O$_2$ kg$^1$ h$^1$). In the case of DEH activity, in OPS, spring wheat species was found have a significant influence ($p \leq 0.05$). The DEH activity was 5% higher in soil samples from *T. sphaerococcum* than in those from *T. persicum*. Wheat species significantly ($p \leq 0.05$ affected soil PER activity in both OPS and CPS. In both production systems, activity was significantly higher in *T. sphaerococcum* compared with *T. persicum*.

**Table 3   The activity of dehydrogenases (DEH), catalase (CAT) and peroxidases (PER) in soil.**

| Sowing density (seeds m$^{-2}$) [B] | OPS | | | CPS | | |
|---|---|---|---|---|---|---|
| | Spring wheat species [A] | | | | | |
| | *T. sphaerococcum* | *T. persicum* | mean | *T. sphaerococcum* | *T. persicum* | mean |
| | DEH (mg TPF kg$^{-1}$ 24h$^{-1}$) | | | | | |
| 400 | $0.476 \pm 0.007$ | $0.426 \pm 0.007$ | $0.451 \pm 0.035$ | $0.625 \pm 0.009$ | $0.621 \pm 0.011$ | $0.623 \pm 0.003$ |
| 500 | $0.482 \pm 0.008$ | $0.470 \pm 0.003$ | $0.476 \pm 0.008$ | $0.651 \pm 0.010$ | $0.640 \pm 0.001$ | $0.646 \pm 0.008$ |
| 600 | $0.497 \pm 0.007$ | $0.489 \pm 0.006$ | $0.493 \pm 0.006$ | $0.684 \pm 0.008$ | $0.650 \pm 0.004$ | $0.667 \pm 0.024$ |
| mean | $0.485 \pm 0.009$ | $0.462 \pm 0.026$ | $0.473 \pm 0.017$ | $0.653 \pm 0.030$ | $0.637 \pm 0.015$ | $0.645 \pm 0.019$ |
| | LSD$_{0.05}$A − 0.021; B − 0.004; A/B − 0.019; B/A − 0.006 | | | LSD$_{0.05}$A − ns[*]; B − 0.013; A/B − 0.049 ; B/A − 0.019 | | |
| | CAT (mg H$_2$O$_2$ kg$^{-1}$ h$^{-1}$) | | | | | |
| 400 | $0.458 \pm 0.009$ | $0.446 \pm 0.005$ | $0.452 \pm 0.008$ | $0.807 \pm 0.008$ | $0.792 \pm 0.004$ | $0.800 \pm 0.011$ |
| 500 | $0.480 \pm 0.013$ | $0.459 \pm 0.013$ | $0.470 \pm 0.015$ | $0.834 \pm 0.006$ | $0.804 \pm 0.004$ | $0.819 \pm 0.021$ |
| 600 | $0.519 \pm 0.004$ | $0.515 \pm 0.008$ | $0.517 \pm 0.003$ | $0.879 \pm 0.011$ | $0.830 \pm 0.006$ | $0.854 \pm 0.035$ |
| mean | $0.485 \pm 0.031$ | $0.473 \pm 0.037$ | $0.479 \pm 0.034$ | $0.840 \pm 0.036$ | $0.809 \pm 0.019$ | $0.824 \pm 0.027$ |
| | LSD$_{0.05}$A − ns; B − 0.025; A/B −ns; B/A − ns | | | LSD$_{0.05}$A − 0.021; B − 0.023; A/B −ns; B/A − ns | | |
| | PER (mM PPG kg$^{-1}$ h$^{-1}$) | | | | | |
| 400 | $1.102 \pm 0.009$ | $0.965 \pm 0.018$ | $1.033 \pm 0.097$ | $1.452 \pm 0.006$ | $1.379 \pm 0.011$ | $1.415 \pm 0.052$ |
| 500 | $1.244 \pm 0.046$ | $1.182 \pm 0.014$ | $1.213 \pm 0.044$ | $1.543 \pm 0.020$ | $1.432 \pm 0.009$ | $1.487 \pm 0.078$ |
| 600 | $1.411 \pm 0.023$ | $1.283 \pm 0.035$ | $1.347 \pm 0.091$ | $1.603 \pm 0.008$ | $1.519 \pm 0.011$ | $1.561 \pm 0.059$ |
| mean | $1.252 \pm 0.155$ | $1.143 \pm 0.162$ | $1.198 \pm 0.158$ | $1.533 \pm 0.076$ | $1.443 \pm 0.071$ | $1.488 \pm 0.073$ |
| | LSD$_{0.05}$A − 0.049; B − 0.084; A/B −ns; B/A − ns | | | LSD$_{0.05}$A − 0.036; B − 0.031; A/B −ns; B/A − ns | | |

**Notes.**

[*]ns - not significant.

Sowing density was demonstrated to have a significant effect on the activity of studied soil enzymes under both production systems for the examined primary wheat species. The highest CAT activity (average 0.854 mg H$_2$O$_2$ kg$^1$ h$^1$), DEH activity (0.667 mg TPF kg$^1$ 24 h$^1$) and PER activity (1.561mM PPG kg$^1$ h$^1$) were found in soil samples with sowing density increased to 600 seeds m$^2$ (CPS). These values were 42%, 27% and 20% respectively, higher relative to OPS values.

The enzyme activity results were used to calculate the soil fertility index. Figure 2 shows the soil G*Mea* values calculated from two primary wheat production systems. Higher G*Mea* values were obtained in soil samples from the CPS experiment (average 0.925). They were 30% higher relative to OPS. It can therefore be concluded that this production system was much more beneficial for the soil for both wheat species. Increasing sowing density caused an increase in soil G*Mea* values, regardless of wheat species. Higher G*Mea* values for soils CPS suggest that the use of this type of production system has a much higher impact on the biological parameters of the soil. This may be due to the lower content of nutrients or their higher availability. According to *García-Ruiz et al. (2008)* the geometric mean of enzyme activities (G*Mea*) index was used to estimate soil quality by changes in soil management.

The calculated correlation coefficient revealed a relationship between clay and the activity of DEH, CAT and PER alike under OPS, at the 0.05 level (Table 4). Based on the

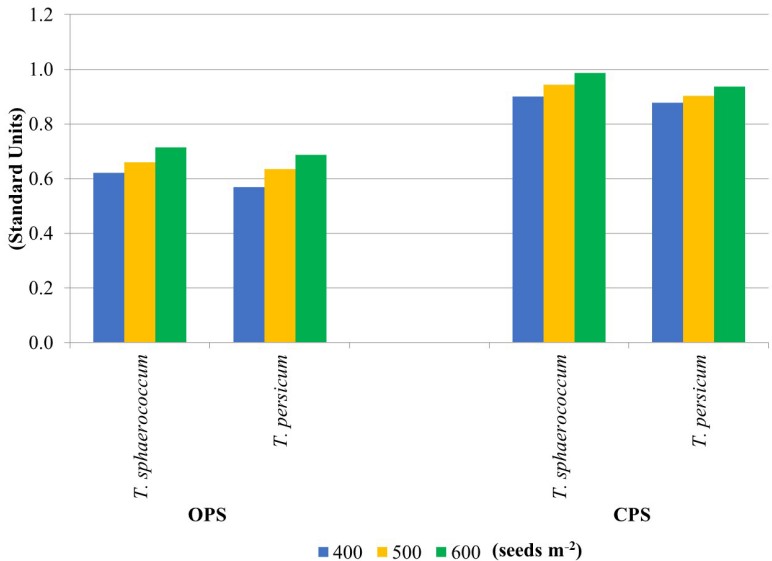

**Figure 2** Soil quality indicator G*Mea*.

coefficient of determination ($R^2$), the influence of clay on the activity of the tested enzymes was only 35%, 31% and 45%, respectively. Similarly, pH significantly affected the activity of CAT ($r = 0.556$, $p \leq 0.05$). The OC content was significantly positively correlated only with PER activity ($r = 0.422$, $p \leq 0.05$), and only in CPS. By contrast, OC did not correlate with DEH or CAT. In OPS, a significant negative correlation was observed between OC and the activity of DEH ($r = -0.623$, $p \leq 0.05$), CAT ($r = -0.843$, $p \leq 0.05$) and PER ($r = -0.778$, $p \leq 0.05$). The coefficient of determination showed that OC had a 39%, 71%, 61%, influence on the activity of the three respective enzymes. Significant positive correlations were also found between the studied soil enzymes in both production systems.

## Phenolic compounds content and antioxidant potential in wheats

The content of TP and CA depended significantly ($p \leq 0.05$) on wheat species and plant density (Table 5). *T. sphaerococcum* plants contained significantly more p-phenolic compounds and CA as compared to *T. persicum* in both CPS and OPS. For the two species (*T. sphaerococcum, T. persicum*), contents of TP were on average 8.71% and 11% higher, and CA contents were 0.6% and 3.4% higher, respectively. Wheat plants sown in a density of 400 seeds m$^2$ had the highest content of the tested compounds, and increasing the density contributed to a reduction in contents. A similar pattern was ascertained for the anti-oxidative potential of FRAP. An average level of anti-oxidative potential of 9.165 mmol Fe$^{2+}$ kg$^{-1}$ DM and 8.736 mmol Fe$^{2+}$ kg$^{-1}$ DM (CPS) was obtained in OPS and CPS, respectively.

**Table 4  The correlation coefficients (r) between the studied characters soil and wheats produced in the OPS.**

| Features | DEH | CAT | PER | Clay | pH | TP | CA | FRAP | OC | Im |
|---|---|---|---|---|---|---|---|---|---|---|
| CAT | 0.752 | | | | | | | | | |
| PER | 0.900 | 0.869 | | | | | | | | |
| Clay | 0.593 | 0.559 | 0.670 | | | | | | | |
| pH | ns | 0.556 | ns | ns | | | | | | |
| TP | −0.516 | −0.580 | −0.648 | ns | −0.855 | | | | | |
| CA | ns | ns | ns | ns | −0.922 | 0.780 | | | | |
| FRAP | ns | −0.681 | −0.622 | ns | −0.875 | 0.858 | 0.711 | | | |
| OC | −0.623 | −0.843 | −0.778 | −0.495 | ns | ns | ns | 0.464 | | |
| Im | −0.599 | ns | −0.521 | ns | −0.530 | 0.732 | 0.531 | 0.429 | ns | |
| L | 0.537 | ns | 0.457 | 0.698 | −0.606 | ns | 0.806 | ns | −0.507 | ns |

**Notes.**

Indicates that the correlation is significant at the 0.05 probability level. >0.413.

Indicates that the correlation is significant at the 0.01 probability level. >0.526.

ns - non-significant.

DEH, dehydrogenases; CAT, catalase; PER, peroxidases; Clay, pH, soil acidity; TP, total polyphenol compounds content; CA, chlorogenic acid content; FRAP, antioxidant potential; OC, organic carbon; Im, Oulema spp. adults; L, Oulema spp. Larvae.

**Table 5  The content of TP, CA and FRAP in plants.**

| Sowing density (seeds m$^{-2}$) [B] | OPS | | | CPS | | |
|---|---|---|---|---|---|---|
| | Spring wheat species [A] | | | | | |
| | *T. sphaerococcum* | *T. persicum* | mean | *T. sphaerococcum* | *T. persicum* | mean |
| | TP ($\mu$g g$^{-1}$ DM) | | | | | |
| 400 | 4492 ± 50.1 | 4323 ± 80.2 | 4408 ± 61.1 | 3744 ± 43.5 | 3588 ± 44.9 | 3666 ± 45.2 |
| 500 | 3759 ± 78.6 | 3133 ± 46.6 | 3446 ± 42.0 | 3414 ± 20.4 | 3220 ± 94.6 | 3317 ± 26.8 |
| 600 | 3509 ± 60.7 | 3136 ± 51.3 | 3323 ± 20.4 | 3270 ± 75.2 | 2789 ± 20.4 | 3030 ± 64.0 |
| mean | 3920 ± 60.2 | 3531 ± 91.6 | 3726 ± 51.1 | 3476 ± 30.9 | 3199 ± 46.8 | 3338 ± 81.1 |
| | LSD$_{0.05}$A − 121.2; B − 160.8; A/B − 191.2; B/A − 227.4 | | | LSD$_{0.05}$A − 96.0; B − 136.2; A/B − 158.1; B/A − 192.6 | | |
| | CA ($\mu$g g$^{-1}$ DM) | | | | | |
| 400 | 1316 ± 1.38 | 1289 ± 1.14 | 1303 ± 14.6 | 1286 ± 8.25 | 1267 ± 2.45 | 1277 ± 12.0 |
| 500 | 1303 ± 1.03 | 1254 ± 1.10 | 1278 ± 26.1 | 1269 ± 1.61 | 1249 ± 0.55 | 1259 ± 10.6 |
| 600 | 1300 ± 0.84 | 1247 ± 2.24 | 1273 ± 28.4 | 1232 ± 8.25 | 1248 ± 2.38 | 1240 ± 10.7 |
| mean | 1306 ± 7.60 | 1263 ± 19.40 | 1285 ± 26.2 | 1262 ± 24.54 | 1255 ± 9.45 | 1258 ± 18.59 |
| | LSD$_{0.05}$A − 3.07; B − 1.97; A/B − 3.54; B/A − 2.79 | | | LSD$_{0.05}$A − 6.49; B - 7.33; A/B − 9.33; B/A − 10.37 | | |
| | FRAP (mmol Fe$^{2+}$ kg$^{-1}$ DM) | | | | | |
| 400 | 9.771 ± 0.063 | 9.297 ± 0.068 | 9.534 ± 0.263 | 9.296 ± 0.071 | 9.050 ± 0.420 | 9.173 ± 0.347 |
| 500 | 9.306 ± 0.042 | 8.981 ± 0.036 | 9.144 ± 0.179 | 8.973 ± 0.110 | 8.764 ± 0.155 | 8.868 ± 0.182 |
| 600 | 8.842 ± 0.073 | 8.794 ± 0.041 | 8.818 ± 0.069 | 8.437 ± 0.087 | 7.898 ± 0.063 | 8.168 ± 0.299 |
| mean | 9.306 ± 0.401 | 9.024 ± 0.223 | 9.165 ± 0.349 | 8.902 ± 0.382 | 8.571 ± 0.580 | 8.736 ± 0.509 |
| | LSD$_{0.05}$A − 0.088; B − 0.092; A/B − 0.121; B/A − 0.130 | | | LSD$_{0.05}$A − 0.256; B − 0.312; A/B −ns[*]; B/A − ns | | |

**Notes.**

[*]ns - not significant.
**Table 6** The correlation coefficients (r) between the studied characters soil and wheats produced in the CPS.

| Features | DEH | CAT | PER | Clay | pH | TP | CA | FRAP | OC | Im |
|---|---|---|---|---|---|---|---|---|---|---|
| CAT | 0.696 | | | | | | | | | |
| PER | 0.523 | 0.926 | | | | | | | | |
| Clay | 0.624 | 0.465 | ns | | | | | | | |
| pH | ns | 0.527 | 0.609 | ns | | | | | | |
| TP | ns | ns | ns | −0.692 | ns | | | | | |
| CA | −0.677 | −0.636 | −0.511 | −0.807 | ns | 0.639 | | | | |
| FRAP | −0.512 | −0.474 | −0.454 | −0.753 | ns | 0.825 | 0.674 | | | |
| OC | ns | ns | 0.422 | ns | ns | −0.721 | ns | −0.533 | | |
| Im | ns | 0.415 | ns | ns | 0.764 | ns | ns | ns | ns | |
| L | ns | 0.577 | 0.670 | ns | 0.599 | ns | ns | ns | ns | ns |

Notes.
Indicates that the correlation is significant at the 0.05 probability level. >0.413.
Indicates that the correlation is significant at the 0.01 probability level. >0.526.
[1]ns - non-significant.

## Relationship between the soil parameters and wheats quality characteristics

The obtained results showed a relationship between the tested soil parameters and wheat quality characteristics. Under OPS (Table 4), increases in the studied soil enzyme activities resulted in a decrease in polyphenolic compound contents in plants. This is evidenced by the significantly negative correlations between TP contents and the activity of the enzymes DEH ($r = -0.516$, $p \leq 0.05$), CAT ($r = -0.580$, $p \leq 0.05$) and PER ($r = -0.648$, $p \leq 0.05$) Meanwhile, in CPS (Table 6), TP content increased because of fell OC content in the soil: OC ($r = -0.721$, $p \leq 0.05$).

Moreover, the activity of the soil enzymes DEH, CAT and PER were also found to have an inverse relationship with CA content and anti-oxidative potential in the tested CPS grown wheats (Table 6). This is confirmed by the negative correlation coefficients for CA: $r = -0.677$ ($p \leq 0.01$), $r = -0.636$ ($p \leq 0.01$), $r = -0.511$ ($p \leq 0.05$), and for anti-oxidative potential: $r = -0.512$ ($p \leq 0.05$), $r = -0.474$ ($p \leq 0.05$), $r = -0.454$ ($p \leq 0.05$), respectively. Additionally, it was noted that among the studied soil enzymes, only the activity of CAT and PER was inversely dependent on the anti-oxidative potential of the wheats cultivated organically: $r = -0.681$ ($p \leq 0.01$), $r = -0.622$ ($p \leq 0.01$) (Table 4).

The abundance of *Oulema* spp. depends on the quality features of soil and wheats. In the flag leaf stage (of *T. sphaerococcum* and *T. persicum*), more *Oulema* spp.—both adults and larvae—were collected from the OPS grown wheat than from the CPS grown wheat (Table 7). Although the number of larvae for OP and CP was not statistically compared, it is clearly visible that in the entomological net there were many times more *Oulema* spp. larvae in OPS than in CPS (57 and 1 no per plot, respectively; no per plot $= 21m^2$). However, in OPS, no significant differences in the number of *Oulema* spp. adults were found between the assayed spring wheat species. The applied sowing density of wheat significantly ($p \leq 0.05$) affected the presence of the pest in the crop. At the flag leaf stage, the density of 500 seeds per m$^2$ proved to be least preferred by adult *Oulema* spp. (Table 7) while in CPS, this was

**Table 7  Insect abundances of *Oulema* spp. adults and larvae.**

| Sowing density (seeds m$^{-2}$) [B] | OPS | | | CPS | | |
|---|---|---|---|---|---|---|
| | Spring wheat species [A] | | | | | |
| | *T. sphaerococcum* | *T. persicum* | mean | *T. sphaerococcum* | *T. persicum* | mean |
| | *Oulema* spp. adults (no per plot) | | | | | |
| 400 | 8.8 ± 0.29 | 11.8 ± 2.22 | 10.3 ± 2.17 | 5.3 ± 0.29 | 2.0 ± 0.41 | 3.6 ± 1.77 |
| 500 | 7.6 ± 0.85 | 5.6 ± 0.75 | 6.6 ± 1.30 | 3.6 ± 0.95 | 5.3 ± 0.65 | 4.4 ± 1.15 |
| 600 | 8.4 ± 1.11 | 6.0 ± 0.71 | 7.2 ± 1.53 | 5.4 ± 0.25 | 3.8 ± 0.50 | 4.6 ± 0.94 |
| mean | 8.3 ± 0.89 | 7.8 ± 3.19 | 8.0 ± 2.31 | 4.8 ± 0.99 | 3.7 ± 1.47 | 4.2 ± 1.34 |
| | LSD$_{0.05}$A − ns$^{*}$; B − 1.04; A/B − 1.43; B/A − 1.47 | | | LSD$_{0.05}$A − 0.46; B − 0.57; A/B − 0.70; B/A − 0.81 | | |
| | *Oulema* spp. larvae (no per plot) | | | | | |
| 400 | 102.8 ± 1.66 | 2.8 ± 0.50 | 52.8 ± 53.46 | 1.0 ± 0.41 | 0.4 ± 0.48 | 0.7 ± 0.53 |
| 500 | 108.8 ± 2.25 | 6.3 ± 0.50 | 57.5 ± 54.81 | 2.3 ± 0.29 | 0.9 ± 0.48 | 1.6 ± 0.82 |
| 600 | 123.0 ± 2.92 | 3.9 ± 1.03 | 63.4 ± 63.71 | 2.1 ± 0.48 | 0.4 ± 0.48 | 1.3 ± 1.04 |
| mean | 111.5 ± 9.12 | 4.3 ± 1.66 | 57.9 ± 55.13 | 1.8 ± 0.69 | 0.5 ± 0.50 | 1.2 ± 0.87 |
| | LSD$_{0.05}$A − 3.16; B − 1.06; A/B − 2.79; B/A − 1.49 | | | LSD$_{0.05}$A − 0.64; B − 0.33; A/B − 0.62; B/A − 0.46 | | |

**Notes.**
$^{*}$ns - not significant.

only observed in *T. sphaerococcum* wheat. On the other hand, *Oulema* spp. larvae were least abundant at the lowest sowing density, *i.e.,* 400 seeds m$^2$, of both wheat species, in both OPS and CPS. It should be emphasised that in neither production system were any constant trends found in the influence of sowing density on abundance of *Oulema* spp.

In the OPS of spring wheat, a high correlation was demonstrated between CA content and the pests, both in the larval and adult form, at $r = 0.806$ and $r = 0.531$, respectively (Table 4). Also, a significant relationship ($p \leq 0.01$) was obtained between TP content and the presence of adult pests ($r = 0.732$). However, the correlation between antioxidant capacity and adults was significant ($r = 0.429$), but at the level of 0.05. No such relationships were obtained in intensive cultivation.

## DISCUSSION

The results of our study showed that CPS contributed to a higher OC content and an increase in soil pH compared to OPS. The study showed differences in enzymatic activity, which results from, among other things, the different agricultural systems (OPS and CPS). Research by *Frac et al. (2011)*, *Kwiatkowski et al. (2020)* and *Kobierski et al. (2020)* showed higher soil enzyme activity in OPS, which is contrary to the results presented here. The course of enzymatic processes in soil is difficult to precisely understand due to the various factors effecting the properties of the soil environment (*Debska et al., 2016*; *Lemanowicz, 2019*). According to *Marinari et al. (2006)*, *Lemanowicz et al. (2020a)* and *Lemanowicz et al. (2020b)* the minimum period for OPS to improve physical, chemical, and biological soil parameters is seven years. Only after this time do the soil properties improve.

Soil enzymatic activity is related to plant species and variety, which is associated with the chemical and microbiological composition of root secretions. Plants influence the accumulation of specific substrates for enzymatic reactions in the soil. The excretions of plant root systems change the physicochemical properties of the soil—especially in the rhizosphere zone, which differs in its species composition of soil microorganisms (*Lemanowicz et al., 2020a*; *Lemanowicz et al., 2020b*; *Lemanowicz & Bartkowiak, 2013*; *Neori et al., 2000*).

*Cenini et al. (2016)* showed a positive correlation between soil enzyme activity and OC content. However, our research found a negative correlation between these parameters. *Burns et al. (2013)*, and *Feng et al. (2018)* found that activating enzyme activity can accelerate the decomposition rate of soil organic matter, leading to a depletion of soil OC. Those authors believe that in arable soils of low soil OC content, enzyme activity can be inhibited by a lack of energy and substrates. This suggests that enzymatic activity did not perfectly reflect SOC content. The PER are enzymes responsible for, among other things, the degradation of lignin, which can be a main component of soil organic matter. This is probably why PER activity correlated positively with OC content, which was greater in CPS than in OPS. The increase in PER activity in CPS is likely related to increased aeration of the crop.

Many authors use multi-parametric indicators to determine soil quality using both enzymatic activity and selected physicochemical parameters (*Debska et al., 2016*; *Wyszkowska et al., 2013*; *Mierzwa-Hersztek et al., 2019*). The geometric mean enzyme activity (G*Mea*) has proven to be a reliable index for estimating soil quality. According to *Paz-Ferreiro et al. (2012)* G*Mea* values are related to soil properties. The G*Mea* calculated in our study showed that the qualitatively best soil was collected from *T. sphaerococcum* at a sowing density of 600 m$^2$ in CPS.

Clay content was demonstrated to affect soil enzyme activity (*An et al., 2015*). Enzymatic activity can be increased, or at least sustained, if adsorption to clay minerals stabilises the enzymes' structure, allowing them to maintain catalytic activity. Changing the concentration of H+ ions can change the concentration of inhibitors or activators in the soil, as well as substrates that directly influence enzyme activity. Each soil enzyme has a specific pH range for optimal activity. At optimal pH, enzymes are more stable. However, at much higher or lower pH levels, soil enzymes are irreversibly denatured and degraded.

Cereals contain numerous antioxidant compounds, including TP. Among TP, ferulic and CA are of particular importance (*Karakaya, 2004*). They are found in all plant parts (*D'Archivio et al., 2007*; *Dai & Mumper, 2010*). The opinion exists that TP contents in wheat depend significantly on genotype (*Lempereur, Rouau & Abecassis, 1997*; *Yu & Zkou, 2004*; *Mikulajová et al., 2007a*; *Mikulajová et al., 2007b*), which is confirmed by our research. The research of other authors (*Klepacka, Gujska & Michalak, 2011*; *Hernández et al., 2011*; *Li, Shewry & Ward, 2008*; *Mazzoncini et al., 2015*) also indicates a relationship between the TP contents and genetically determined cereal crop characteristics.

In addition to genetic factors, environmental factors too may play an important role in the quality and quantity of biosynthesis of TP (*Stumpf, Yan & Honermeier, 2019*; *Przybylska-Balcerek, Frankowski & Stuper-Szablewska, 2020*). According to *Lu et al. (2015)*,

the composition of bioactive compounds is much more influenced by environmental conditions (E) than by genotype (G). They evidenced that the magnitude of the influence of E was greater for total TP (43%) than for phenolic acid contents (40%). The environmental factors include abiotic stress, nutrient availability, climate, water supply, and other growth conditions such as plant density (*Stumpf, Yan & Honermeier, 2019*; *Przybylska-Balcerek, Frankowski & Stuper-Szablewska, 2020*). Soils with a lower carbon content contain less humus, which makes them low in nutrients (macro elements). In such soils, plants are more exposed to abiotic stress, which causes an increase in the content of polyphenolic compounds. This result is consistent with what has been stated and documented by *Boscaiu et al. (2010)*, *Bautista et al. (2016)* and *Yang et al. (2018)*, who found that conditions of abiotic stress generated changes in metabolic processes (*e.g.*, the shikimate pathway and phenylalanine) and induced an increase in the concentration of phenolic compounds. These authors also noted that under natural conditions, there were usually combinations of different stress factors that induced response mechanisms, such as physiological tolerance (biosynthesis of compounds) or evasion before death.

Each crop system and genotype are believed to have its own specific, preferred density. Plant density recommendations should also consider the specific environmental conditions (*Royo et al., 2006*; *Xihuan, Wensuo & Caiying, 2010*). Cereal crops quality can be maximised when competition between them is minimised, as this allows the plant to maximise its exploitation of resources. The appropriate density of wheat per unit area results in a better use of soil nutrients (*Kheshtzar & Siadat, 2015*; *Hiltbrunner, Streit & Lidgens, 2007*). Excessively high-density planting leads to a deficiency of soil nutrients, especially nitrogen. Nitrogen deficiency also occurs in OPS soils because low organic matter means low plant-available N rates too since organic matter is a natural source of N to crops. A lower nitrogen supply to plants increases the concentration of TP in vegetables and medicinal plants (*Sousa et al., 2008*; *Giorgi et al., 2009*). In our research, wheat contained lower contents of TP when grown in the OPS than in CPS. The same was observed by *Cartelat et al. (2005)*, whose research was consistent with this study, and especially relating to wheat leaf tissue. Some studies indicated that wheat plants density contributed to the deterioration in crop quality (*Mosanaei et al., 2017*). This study also confirmed this relationship with regard to bioactive compounds. The demonstrated strong relationship between the anti-oxidative activity and TP contents in both CPS and OPS wheat production indicates their significant contribution to the anti-oxidative activity of cereals. As in our results, *Mikulajová et al. (2007b)* showed high values of the relationship between anti-oxidative activity measured by DPPH, ABTS and EPR and phenols at the level of $R = 0.958, 0.907$ and $0.934$, respectively. The OPS grown wheat shows increased anti-oxidative activity compared to CPS grown wheat (*Mazzoncini et al., 2015*).

The TP leached from green leaves—and root secretions to the soil—affect the rate of organic matter decomposition, playing a key role in the circulation of nutrients (*Min et al., 2015*). In soil, TP are broken down by enzymes (phenol oxidase or PER) secreted by fungi and bacteria (*Sinsabaugh, 2010*). However, the relationship between TP contents and soil enzyme activity is not clear. The interpretation of this relationship depends on whether TP are a product of, or substrate for, soil enzymes. The negative correlations with

DEH, CAT and PER activity obtained in this study indicate that the TP were probably acting as a substrate. Long-term cultivation of plants of high allelopathic potential may result in relatively high contents of TP accumulating in the soil, reducing the abundance of microorganisms and thereby reducing the enzymatic activity of the soil. The study also found a negative relationship between TP contents in wheat and OC content. Research by *Mallik (1997)* showed that the accumulation of organic matter was lower in soils changed by various TP than in an unchanged control soil.

The most important cereal pest in Poland is *Oulema* ssp. (*Lamparski, 2016*; *Lamparski et al., 2017*). Larvae feeding on cereal leaves can reduce the assimilation surface of flag and subflag leaves by 50%, and sometimes even 80% (*Kaniuczak & Bereś, 2011*; *Lamparski et al., 2017*). Apart from the volatile organic compounds (VOCs) emitted by plants, secondary metabolites with allelopathic activity also influence the physiology and behaviour of all insects. The TP are the most active allelochemicals (*Mallikarjuna et al., 2004*; *Cipollini et al., 2008*; *Lamparski et al., 2015*). The lower numbers of *Oulema* spp. adults and larvae collected in our study in the flag-leaf stage of *T. sphaerococcum* and *T. persicum* wheat grown CPS, rather than OPS, may result from the lower content of these compounds in plants. *Kaniuczak & Bereś (2011)* state, however, that such a result is mainly due to OPS restricting the use of the full range of methods available for regulating the number of phytophages feeding on plants. Conversely, *Eleftherianos et al. (2006)* showed an inverse correlation between the TP contents and the fertility of insects. It should be added, however, that their experiment was conducted on maize and barley, and the investigated insect was corn aphid and *Rhopalosiphum padi*. Additionally, *Lamparski et al. (2015)* showed a lack of strong relationships between TP contents in wheat and the parameters of insects feeding on crops treated organically. This is indicated by the weak or non-significant correlation coefficients obtained in their research.

According to *Kaniuczak & Bereś (2011)* and *Lamparski et al. (2017)*, there is an absolute need to take actions to reduce pest populations in cereals. Their presence not only contributes to direct losses in crops but through their damage to the above-ground parts of plant tissues. They also facilitate the penetration of fungi, bacteria, and viruses. This is the cause of many diseases occurring that further reduce the quantity and quality of the harvested material. To protect cereal crops against the negative effects of pests many methods are used, ranging from selecting the appropriate variety, environmental conditions, and cultivation technology –in which sowing density is an essential element (*Szczepanek et al., 2020*; *Lemanowicz et al., 2020a*; *Lemanowicz et al., 2020b*).

## CONCLUSIONS

The highest density of plants (600 seeds $m^2$) leads to the highest soil enzymatic activity, both in organic and conventional agriculture. With *T. sphaerococcum* having a more beneficial effect in this regard. Soil in the organic system was characterized by lower enyzmatic activity compared to conventional farming. Despite the lower activity of soil enzymes in the organic system, wheat grown on these soils contained more total polyphenolic compounds. Thus, it has been proven that abiotic factors have the greatest influence on the concentration of

total polyphenolic compounds in plants. Therefore, it is recommended to grow primary wheat in an organic system.

The cultivation system and the total polyphenol content of wheat have a direct effect on the occurrence and population of *Oulema* ssp.. Wheat grown under the organic system was more inhabited by adults, as well as larvae of *Oulema* spp. at one of the main stages of wheat development (flag leaf). The lowest incidence of adults of *Oulema* ssp. on *T. sphaerococcum* was obtained at a sowing density of 500 seeds m$^2$, regardless of the spring wheat tillage system used. On the other hand, the incidence of larvae of this pest was the lowest at a sowing density of 400 seeds m$^2$. Therefore, the problem of colonization of primary wheat by *Oulema* ssp. in agroecology should be subject to further scientific research.

### Funding
The publication was financed from the European Union funds under the Cooperation of the Rural Development Programme for 2014-2020 (63.63%) and National Public Funds (36.37%). The Managing Authority of the Rural Development Programme for 2014-2020 the Minister of Agriculture and Rural Development and the European Agricultural Fund for Rural Development: Europe investing in rural areas. The funders had no role in study design, data collection and analysis, decision to publish, or preparation of the manuscript.

### Grant Disclosures
The following grant information was disclosed by the authors:
European Union funds under the Cooperation of the Rural Development Programme for 2014-2020 (63.63%).
National Public Funds (36.37%).
Minister of Agriculture and Rural Development.
European Agricultural Fund for Rural Development: Europe investing in rural areas.

### Competing Interests
The authors declare there are no competing interests.

### Author Contributions
- Jarosław Pobereżny performed the experiments, analyzed the data, prepared figures and/or tables, authored or reviewed drafts of the article, and approved the final draft.
- Elżbieta Wszelaczyńska performed the experiments, analyzed the data, prepared figures and/or tables, authored or reviewed drafts of the article, and approved the final draft.
- Robert Lamparski performed the experiments, analyzed the data, prepared figures and/or tables, authored or reviewed drafts of the article, and approved the final draft.
- Joanna Lemanowicz conceived and designed the experiments, performed the experiments, analyzed the data, prepared figures and/or tables, authored or reviewed drafts of the article, and approved the final draft.
- Agata Bartkowiak performed the experiments, analyzed the data, prepared figures and/or tables, authored or reviewed drafts of the article, and approved the final draft.

- Małgorzata Szczepanek conceived and designed the experiments, performed the experiments, analyzed the data, prepared figures and/or tables, authored or reviewed drafts of the article, and approved the final draft.
- Katarzyna Gościnna performed the experiments, analyzed the data, prepared figures and/or tables, authored or reviewed drafts of the article, and approved the final draft.

## Data Availability

The raw measurements are available as a Supplemental File.

## Supplemental Information

Supplemental information for this article can be found online at http://dx.doi.org/10.7717/peerj.14916#supplemental-information.

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
