# Peer review of "The impact of spring wheat species and sowing density on soil biochemical properties, content of secondary plant metabolites and the presence of Oulema ssp"

_PeerJ, doi:10.7717/peerj.14916_

## Round 0.1 · original submission · Minor Revisions

Dear authors,

All reviewers recommended minor revisions on this manuscript. Please revise carefully according to the comments of reviewers and respond questions/comments one by one.

Thanks for all your efforts made on this manuscript. We are looking for your revised version.

·

Basic reporting

Main issues:
1. It is suggested to introduce the progress and result by sticking on a consistent logical line for the designing of the experiments and the major discoveries. For example, in the Abstract, Line 47-50, it may be more reasonable to introduce like “To the best of our knowledge, the studies conducted so far do not sufficiently reveal the impacts of the wheat species and the sowing density, together with the biochemical properties of the soil, on the accumulation of bioactive ingredients in the crop plants, and the subsequent impacts on the occurrence of phytophagic entomofauna in various management systems.”.
2. It is suggested to describe the Results under several Sections according to the type of the analyses performed.
3. For the significance analyses in the Results, please indicate the significance level with labels in Tables 2, 3, 4, and 5.
4. For the plant sampling in the Materials and Methods, please consider describing in detail the standard for sample collection, such as tissue types, growing stages, spatial sites in a plot, and cleaning methods.
5. Please pay attention to the use of the abbreviations. Generally, the full version should be mentioned where first emerge in the text and thereafter only the short version should be used, such as “organic carbon (OC)”, “Chlorogenic Acid (CA)”, and “antioxidant capacity (FRAP)”.
6. The language of the text needs improvement, there were some mistakes in spelling, phrases, styles, and grammar. For example, in the Abstract, Line 65-67, does it mean “Regardless of the production system, the occurrence of the Oulema spp. adults on T. sphaerococcum was the lowest at a sowing density of 500 seeds m-2, and the occurrence of the larvae was the lowest at a sowing density of 400 seeds m-2.” I listed several of them below, and the manuscript needs focused, discreet proofreading.



Some other issues:

1. Abstract, Line 54, “…systems.” Line 57, “…densities…”. Line 62, “…grown in OPS.”
2. In Table 6, please double check the legend.
3. In Figure 2, please indicate the units for the Y-axis values. Please also indicate the seed densities with units.
4. Line 105, “… phenolic acids...”.
5. Line 152, “…for the wheat species were…”
6. Line 154, “The sowing amounts…”.
7. Line 164-166, “…, the soil pH in CaCl2 (0.01 M) was measured …, OC was measured using…”.
8. Line 171-172, “… at 546 nm was carried out…”.

Experimental design

Good

Validity of the findings

Good

Reviewer 2 ·

Basic reporting

1. The measure units were not included in all the tables and supplementary table, please add it.
2. Figure1 was not cited in the manuscript.

Experimental design

no comments

Validity of the findings

There is no statistical analysis on table 1, please include it.

Additional comments

In the article “The impact of spring wheat species and sowing density on soil biochemical properties, content of secondary plant metabolites and the presence of Oulema spp. ”,the authors brings to readers the effect of wheat species and sowing density on soil properties and plant biological compounds and insect pests in organic and conventional farming systems, which gave us more information to sustainable agriculture. However, I still have comes comments.

1. Did the author investigate the effect of sow density, and wheat species on the yield?
2. Line 310, the author claimed that “polyphenol content fell because of increased OC content in the soil: OC ”, please provide solid evidence that can prove the causation.
3. Line 322, is there any difference between the larvae between OPS and CPS?

Reviewer 3 ·

Basic reporting

Please see attached PDF for edits.

Experimental design

Great study with interesting results.

Validity of the findings

no comment

Additional comments

Overall, well done.

Annotated reviews are not available for download in order to protect the identity of reviewers who chose to remain anonymous.

---

## Round 0.2 · Minor Revisions

Please make minor revisions based on the 2nd reviewer's comments. Thanks!

·

Basic reporting

Meets Publish Standard and Recommend to Accept

Experimental design

Meets Publish Standard and Recommend to Accept

Validity of the findings

Meets Publish Standard and Recommend to Accept

Additional comments

Meets Publish Standard and Recommend to Accept

Reviewer 2 ·

Basic reporting

no comments

Experimental design

no comments

Validity of the findings

no comments

Additional comments

Minor comments

1. The authors used FRAP as abbreviation for antioxidant capacity for several times, please correct it.
2. Please follow the same units of measurement in methods part, eg. either “milliliter” or mL.

---

## Round 0.3 · Minor Revisions

Thank you so much for all your contributions during the reviewing process. I am glad you have addressed all of the reviewers' comments. However, the conclusion still needs to be improved to meet the publication requirements. Following are the comments from our Section Editor,

> Please review your conclusions. Actually, the actual conclusion is only a selection of some results. The use of many acronyms makes the conclusions difficult to read. Please put your results in the context of current knowledge and make clear your scientific contribution. Did you discover an interesting phenomenon, or do you suggest any mechanism (e.g. in agroecology) that could be further investigated or applied in agriculture?'

Please carefully revise the Conclusions according to Section Editor's comments. I am looking forward to your updated manuscript. Thanks.

---

## Round 0.4 · accepted · Accept

Thanks for the authors' careful revision. I think you have improved the conclusions based on the section editor's comments. Congratulations!

When approving the decision, the Section Editor mentioned:
> I would remove the 'In this study' (first sentence of Conclusions); it's redundant. "In this study, our results provide a...."